# Algorithmic Linearly Constrained Gaussian Processes

**Markus Lange-Hegermann**
Department of Electrical Engineering and Computer Science
Ostwestfalen-Lippe University of Applied Sciences
Lemgo
`markus.lange-hegermann@hs-owl.de`

## Abstract

We algorithmically construct multi-output Gaussian process priors which satisfy linear differential equations. Our approach attempts to parametrize all solutions of the equations using Gröbner bases. If successful, a push forward Gaussian process along the paramerization is the desired prior. We consider several examples from physics, geomathematics and control, among them the full inhomogeneous system of Maxwell's equations. By bringing together stochastic learning and computer algebra in a novel way, we combine noisy observations with precise algebraic computations.

## 1 Introduction

In recent years, Gaussian process regression has become a prime regression technique [37]. Roughly, a Gaussian process can be viewed as a suitable[1] probability distribution on a set of functions, which we can condition on observations using Bayes' rule. The resulting mean function is used for regression. The strength of Gaussian process regression lies in *avoiding overfitting* while still finding functions complex enough to describe *any behavior* present in given observations, even in noisy or unstructured data. Gaussian processes are usually applied when observations are rare or expensive to produce. Applications range, among many others, from robotics [9], biology [19], global optimization [33], astrophysics [13] to engineering [47].

Incorporating justified assumptions into the prior helps these applications: the full information content of the scarce observations can be utilized to create a more precise regression model. Examples of such assumptions are smooth or rough behavior, trends, homogeneous or heterogeneous noise, local or global behavior, and periodicity (cf. §4 in [37],[11]). Such assumptions are usually incorporated in the covariance structure of the Gaussian process.

Even certain physical laws, given by certain linear differential equations, could be incorporated into the covariance structures of Gaussian process priors. Thereby, despite their random nature, all realizations and the mean function of the posterior strictly adhere to these physical laws[2]. For example, [29, 41] constructed covariance structures for divergence-free and curl-free vector fields, which [50, 45] used to model electromagnetic phenomena.

A first step towards systematizing this construction was achieved in [24]. In certain cases, a map into the solution set for physical laws could be found by a computation that does not necessarily terminate. Having found such a map, one could assume a Gaussian process prior in its domain and push it forward. This results in a Gaussian process prior for the solutions of the physical laws.

In Section 2, we stress that the map from [24] into the solution set should be a parametrization, i.e., surjective. In Section 3, we combine this with an algorithm which computes this parametrization if it exists or reports failure if it does not exist.

This algorithm is a homological result in algebraic system theory (cf. §7.(25) in [32]). Using Gröbner bases, it is fully algorithmic and works for a wide variety of operator rings; among them the polynomial ring in the partial derivatives, which models linear systems of differential equations with constant coefficients; the (various) Weyl algebras which model linear systems of differential equations with variable coefficients (cf. Example 4.2); and similar rings for difference equations and combined delay differential equations. To demonstrate the use of Gröbner bases, Example 4.3 shows explicit computer algebra code.

Using the results of this paper, one can add information to Gaussian processes[3] not only by

(i) conditioning on observations (Bayes' rule), but also by

(ii) restricting to solutions of linear operator matrices by constructing a suitable prior.

Since these two constructions are compatible, we can combine *strict, global information* from equations with *noisy, local information* from observations. The author views this combination of techniques from homological algebra and machine learning as the main result of this paper, and the construction of covariance functions satisfying physical laws as a proof of concept.

Even though Gaussian processes are a highly precise interpolation tool, they lack in two regards: missing extrapolation capabilities and high computation time, cubically in the amount of observations. These problems have, to a certain degree, been addressed: more powerfull covariance structures [25, 21, 51, 53, 6] and several fast approximations to Gaussian process regression [48, 18, 52, 20, 10] have been proposed. This paper addresses these two problems from a complementary angle. The linear differential equations allow to extrapolate and reduce the needed amount of observations, which improves computation time.

The promises in this introduction are demonted in Example 4.1. It constructs a Gaussian process such that all of its realizations satisfy the inhomogeneous Maxwell equations of electromagnetism. Conditioning this Gaussian process on a *single* observation of electric current yields, as expected, a magnetic field circling around this electric current. This shows how to combine data (the electric current) with differential equations for a global model, which extrapolates away from the data.

## 2 Differential Equations and Gaussian Processes

This section is mostly expository and summarizes Gaussian processes and how differential operators act on them. Subsection 2.1 summarizes Gaussian process regression. We then introduce differential (Subsection 2.2) and other operators (Subsection 2.3).

### 2.1 Gaussian processes

A *Gaussian process* $g = \mathcal{GP}(\mu, k)$ is a distribution on the set of functions $\mathbb{R}^d \to \mathbb{R}^\ell$ such that the function values $g(x_1), \ldots, g(x_n)$ at $x_1, \ldots, x_n \in \mathbb{R}^d$ have a joint Gaussian distribution. It is specified by a *mean function*

$$\mu : \mathbb{R}^d \to \mathbb{R}^\ell : x \mapsto E(g(x))$$

and a positive semidefinite *covariance function*

$$k : \mathbb{R}^d \oplus \mathbb{R}^d \to \mathbb{R}^{\ell \times \ell}_{\succeq 0} : (x, x') \mapsto E\left((g(x) - \mu(x))(g(x') - \mu(x'))^T\right) .$$

Assume the regression model $y_i = g(x_i)$ and condition on $n$ observations

$$\left\{ (x_i, y_i) \in \mathbb{R}^{1 \times d} \oplus \mathbb{R}^{1 \times \ell} \mid i = 1, \ldots, n \right\} .$$

Denote by $k(x, X) \in \mathbb{R}^{\ell \times \ell n}$ resp. $k(X, X) \in \mathbb{R}^{\ell n \times \ell n}_{\succeq 0}$ the (covariance) matrices obtained by concatenating the matrices $k(x, x_j)$ resp. the positive semidefinite block partitioned matrix with blocks

$k(x_i, x_j)$. Write $\mu(X) \in \mathbb{R}^{\ell \times n}$ for the matrix obtained by concatenating the vectors $\mu(x_i)$ and $y \in \mathbb{R}^{1 \times \ell n}$ for the row vector obtained by concatenating the rows $y_i$. The posterior

$$\mathcal{GP} \Big( \quad x \mapsto \mu(x) + (y - \mu(X))k(X, X)^{-1}k(x, X)^T, $$

$$(x, x') \mapsto k(x, x') - k(x, X)k(X, X)^{-1}k(x', X)^T \Big) ,$$

is again a Gaussian process and its mean function is used as regression model.

## 2.2 Differential equations

Roughly speaking, Gaussian processes are the linear objects among stochastic processes. Hence, we find a rich interplay of Gaussian processes and linear operators.

For simplicity, let $R = \mathbb{R}[\partial_{x_1}, \dots, \partial_{x_d}]$ be the polynomial ring in the partial differential operators. For different or more general operator rings see Subsection 2.3. This ring models linear (partial) differential equations with constant coefficients, as it acts on the vector space $\mathcal{F} = C^\infty(\mathbb{R}^d, \mathbb{R})$ of smooth functions, where $\partial_{x_i}$ acts by partial derivative w.r.t. $x_i$. The set of realizations of a Gaussian process with squared exponential covariance function is dense in $\mathcal{F}$ (cf. Thm. 12, Prop. 42 in [43]).

The class of Gaussian processes is closed under matrices $B \in R^{\ell \times \ell''}$ of linear differential operators with constant coefficients. Let $g = \mathcal{GP}(\mu, k)$ be a Gaussian process with realizations in a space $\mathcal{F}^{\ell''}$ of vectors with functions in $\mathcal{F}$ as entries. Define the Gaussian process $B_*g$ as the Gaussian process induced by the pushforward measure under $B$ of the Gaussian measure induced by $g$. It holds that

$$B_*g = \mathcal{GP}(B\mu(x), Bk(x, x')(B')^T) , \tag{1}$$

where $B'$ denotes the operation of $B$ on functions with argument $x' \in \mathbb{R}^d$ [4, Thm. 9].

The covariance function $k$ for such Gaussian processes $B_*g$ as in (1) is often singular. This is to be expected, as $B_*g$ is rarely dense in $\mathcal{F}^\ell$. For numerical stability, we tacitly assume the model $y_i = g(x_i) + \varepsilon$ for small Gaussian white noise term $\varepsilon$ and adopt $k$ by adding $\mathrm{var}(\varepsilon)$ to $k(x_i, x_i)$ for observations $x_i$.

**Example 2.1.** Let $g = \mathcal{GP}(0, k(x, x'))$ be a scalar univariate Gaussian process with differentiable realizations. Then, the Gaussian process of derivatives of functions is given by

$$\left[\tfrac{\partial}{\partial x}\right]_* g = \mathcal{GP}\left(0, \frac{\partial^2}{\partial x \partial x'}k(x, x')\right) .$$

One can interpret this Gaussian process $\left[\tfrac{\partial}{\partial x}\right]_* g$ as taking derivatives as measurement data and producing a regression model of derivatives.

We say that a Gaussian process is *in* a function space, if its realizations are contained in said space. For $A \in R^{\ell' \times \ell}$ define the *solution set*

$$\mathrm{sol}_\mathcal{F}(A) := \{f \in \mathcal{F}^\ell \mid Af = 0\} .$$

Such solution sets and Gaussian processes are connected in an almost tautological way.

**Lemma 2.2.** *Let $g = \mathcal{GP}(\mu, k)$ be a Gaussian process in $\mathcal{F}^{\ell \times 1}$. Then $g$ is also a Gaussian process in $\mathrm{sol}_\mathcal{F}(A)$ for $A \in R^{\ell' \times \ell}$ if and only if $\mu \in \mathrm{sol}_\mathcal{F}(A)$ and $A_*(g - \mu)$ is the constant zero process.*

*Proof.* Assume that $g$ is a Gaussian process in $\mathrm{sol}_\mathcal{F}(A)$. Then, the mean function is a realization, thus $\mu \in \mathrm{sol}_\mathcal{F}(A)$. Furthermore, for $\tilde{g} := (g - \mu) = \mathcal{GP}(0, k)$ all realizations are annihilated by $A$, and hence $A_*\tilde{g}$ is the constant zero process.

Conversely, assume that $\mu \in \mathrm{sol}_\mathcal{F}(A)$ and $A_*(g - \mu)$ is the constant zero process. This implies $0 = A_*(g - \mu) = A_*g - A_*\mu = A_*g$, i.e. all realizations of $g$ become zero after a pushforward by $A$. In particular, all realizations of $g$ are contained in $\mathrm{sol}_F(A)$. $\square$

This lemma implies another advantage of choosing a zero mean function: it is always a solution of the linear differential equations.

Our goal is to construct Gaussian processes with realizations dense in the solution set $\mathrm{sol}_{\mathcal{F}}(A)$ of an operator matrix $A \in R^{\ell' \times \ell}$. The following remark, implicit in [24], is a first step towards an answer.

**Remark 2.3.** Let $A \in R^{\ell' \times \ell}$ and $B \in R^{\ell \times \ell''}$ with $AB = 0$. Let $g = \mathcal{GP}(0, k)$ be a Gaussian process in $\mathcal{F}^{\ell''}$. Then, the set of realizations of $B_* g$ is contained in $\mathrm{sol}_{\mathcal{F}}(A)$.

This remark is implied by Lemma 2.2, as $A_*(B_* g) = (AB)_* g = 0_* g = 0$.

We call $B \in R^{\ell \times \ell''}$ a *parametrization* of $\mathrm{sol}_{\mathcal{F}}(A)$ if $\mathrm{sol}_{\mathcal{F}}(A) = B\mathcal{F}^{\ell''}$. Parametrizations yield the denseness of the realizations of a Gaussian process $B_* g$ in $\mathrm{sol}_{\mathcal{F}}(A)$.

**Proposition 2.4.** *Let $B \in R^{\ell \times \ell''}$ be a parametrization of $\mathrm{sol}_{\mathcal{F}}(A)$ for $A \in R^{\ell' \times \ell}$. Let $g = \mathcal{GP}(0, k)$ be a Gaussian process dense in $\mathcal{F}^{\ell''}$. Then, the set of realizations of $B_* g$ is dense in $\mathrm{sol}_{\mathcal{F}}(A)$.*

This proposition is a consequence of partial derivatives being bounded, and hence continuous, when $\mathcal{F}$ is equipped with the Fréchet topology generated by the family of seminorms

$$\|f\|_{a,b} := \sup_{\substack{i \in \mathbb{Z}_{\geq 0}^d \\ |i| \leq a}} \sup_{z \in [-b,b]^d} |\frac{\partial}{\partial z^i} f(z)|$$

for $a, b \in \mathbb{Z}_{\geq 0}$ (cf. §10 in [49]). Now, the continuous surjective map induced by $B$ maps a dense set to a dense set.

## 2.3 Further operator rings

The theory presented for differential equations with constant coefficients also holds for other rings $R$ of linear operators and function spaces $\mathcal{F}$. The following three operator rings are prominent examples.

The polynomial ring $R = \mathbb{R}[x_1, \ldots, x_d]$ models polynomial equations when it acts on the set $\mathcal{F}$ of smooth functions defined on a (Zariski-)open set in $\mathbb{R}^d$.

For ordinary linear differential equations with rational[4] coefficients consider the Weyl algebra $R = \mathbb{R}(t)\langle \partial_t \rangle$, with the non-commutative relation $\partial_t t = t\partial_t + 1$ representing the product rule of differentiation. We consider solutions in the set $\mathcal{F}$ of smooth functions defined on a co-finite set.

The polynomial ring $R = \mathbb{R}[\sigma_{x_1}, \ldots, \sigma_{x_d}]$ models linear shift equations with constant coefficients when it acts on the set $\mathcal{F} = \mathbb{R}^{\mathbb{Z}_{\geq 0}^d}$ of $d$-dimensional sequences by translation of the arguments.

# 3 Computing parametrizations

By the last section, constructing a parametrization $B$ of $\mathrm{sol}_{\mathcal{F}}(A)$ yields a Gaussian process dense in the solution set $\mathrm{sol}_{\mathcal{F}}(A)$ of an operator matrix $A \in R^{\ell' \times \ell}$. Subsection 3.1 gives necessary and sufficient conditions for a parametrization to exist and Subsection 3.2 describes their computation.

## 3.1 Existence of parametrizations

It turns out that we can decide whether a parametrization exists purely algebraically, only using operations over $R$ that do not involve $\mathcal{F}$.

By r-ker$(A)$ we denote the right kernel of $A \in R^{\ell' \times \ell}$, i.e. r-ker$(A) = \{m \in R^{\ell \times 1} \mid Am = 0\}$. By l-ker$(A)$ we denote the left kernel of $A$, i.e. l-ker$(A) = \{m \in R^{1 \times \ell'} \mid mA = 0\}$. Abusing notation, denote any matrix as left resp. right kernel if its rows resp. columns generate the kernel as an $R$-module.

**Theorem 3.1.** *Let $A \in R^{\ell' \times \ell}$. Define matrices $B = \text{r-ker}(A)$ and $A' = \text{l-ker}(B)$. Then $\mathrm{sol}_{\mathcal{F}}(A')$ is the largest subset of $\mathrm{sol}_{\mathcal{F}}(A)$ that is parametrizable and $B$ parametrizes $\mathrm{sol}_{\mathcal{F}}(A')$.*

A well-known special case of this theorem are finite dimensional vector spaces, with $R = \mathcal{F}$ a field. In that case, $\mathrm{sol}_{\mathcal{F}}(A)$ can be found by solving the homogeneous system of linear equations $Ab = 0$ with the Gaussian algorithm and write a base for the solutions of $b$ in the columns of a matrix $B$. This matrix $B$ is also called the (right) kernel of $A$. Now, we wonder whether there are additional equations satisfied by the above solutions, i.e. when does $Ab = 0$ imply $A'b = 0$. These equations $A'$ are the (left) kernel of $B$. At least in the case of finite dimensional vector spaces[5], there are no additional equations[6]. However, for general rings $R$, the left kernel $A'$ of the right kernel $B$ of $A$ is not necessarily $A$ (up to an equivalence). For example, the solution set $\mathrm{sol}_{\mathcal{F}}(A')$ is the subset of controllable behaviors in $\mathrm{sol}_{\mathcal{F}}(A)$.

**Corollary 3.2.** *In Theorem 3.1, $\mathrm{sol}_{\mathcal{F}}(A)$ is parametrizable if and only if the rows of $A$ and $A'$ generate the same row-module. Since $AB = 0$, this is the case if all rows of $A'$ are contained in the row module generated by the rows of $A$. In this case, $\mathrm{sol}_{\mathcal{F}}(A)$ is parametrized by $B$. Furthermore, a Gaussian process $g$ with realizations dense in $\mathcal{F}^{\ell''}$ leads to a Gaussian process $B_* g$ with realizations dense in $\mathrm{sol}_{\mathcal{F}}(A)$.*

For a formal proof of this theorem and its corollary see [55, Thm. 2], [54, Thm. 3, Alg. 1, Lemma 1.2.3], or [32, §7.(24)] and for additional characterizations, generalizations, and proofs using more homological machinery see [36, 35, 2, 42, 7, 40] and references therein.

The approach assigns a prior to the parametrising functions and pushes this prior forward to a prior of the solution set $\mathrm{sol}_{\mathcal{F}}(A)$. The paramerization is not canonical, and hence different parametrizations might lead to different priors.

## 3.2 Algorithms

Summarizing Theorem 3.1 and Corollary 3.2 algorithmically, we need to compute right kernels (of $A$), compute left kernels (of $B$), and decide whether rows (of $A'$) are contained in a row module (generated by the rows of $A$). All these computations are an application of Gröbner basis algorithms.

In the recent decades, Gröbner bases algorithms have become one of the core algorithms of computer algebra, with manifold applications in geometry, system theory, natural sciences, automatic theorem proving, post-quantum cryptography, and many others. Reduced Gröbner bases generalize the reduced echelon form from linear systems to systems of polynomial (and hence linear operator) equations, by bringing them into a standard form[7]. They are computed by Buchberger's algorithm, which is a generalization of the Gaussian and Euclidean algorithm and a special case of the Knuth-Bendix completion algorithm.

Similar to the reduced echelon form, Gröbner bases allow to compute all solutions over $R$ (not $\mathcal{F}$) of the homogeneous system and compute, if it exists, a particular solution over $R$ (not $\mathcal{F}$) for an inhomogeneous system. Solving homogeneous systems is the same as computing its right resp. left kernel. Solving inhomogeneous equations decides whether an element is contained in a module. Alternatively, the uniqueness of reduced Gröbner bases also decides submodule equality.

A formal description of Gröbner bases would exceed the scope of this note. Instead, we refer to the excellent literature [46, 12, 1, 17, 14, 5]. Gröbner basis algorithms exist for many rings $R$. They historically emerged from polynomial rings, and have since been generalized to the Weyl algebra, the shift algebra, and, more generally, $G$-algebras [26, 27] and Ore-algebras [39, 38]. They are implemented in various computer algebra systems, Singular [8] and Macaulay2 [16] are two well-known examples. Even though the complexity of Gröbner bases is in the vicinity of EXPSPACE completeness (cf. [30, 31, 3]), the "average interesting example" (e.g. every example in this paper) usually terminates instantaneously. This holds in particular since the Gröbner basis computations only involve the operator equations, but not the data in any way.

### 3.3 Hyperparameters

Many covariance functions[8] incorporate hyperparameters and advanced methods specifically add more hyperparameters to Gaussian processes, see e.g. [44, 6, 51], for additional flexibility. The approach in this paper is the opposite by restricting the Gaussian process prior to solutions of an operator matrix. Of course, the prior of the parametrizing functions can still contain hyperparameters, which can be determined by maximizing the likelihood. Many important applications contain unknown parameters in the equations. Such parameters can also be estimated by the likelihood.

Consider ordinary differential equations, with constant resp. variable coefficients. The solution set of an operator matrix is a direct sum of parametrizable functions and a finite dimensional set of functions, due to the Smith form resp. Jacobson form. In many cases, in particular the case of constant coefficients, the solution set of the finite dimensional summand can easily be computed. This paper also allows to compute with the parametrizable summand of the solution set and estimate parameters and hyperparameters of both summands together.

## 4 Examples

**Example 4.1.** Maxwell's equations of electromagnetism uses curl and divergence operators as building blocks. It is a well-known result that the solutions of the *inhomogeneous* Maxwell equations are parametrized by the electric and magnetic potentials. We verify this and use the parametrization to construct a Gaussian process, such that its realizations adhere to Maxwell's equations. In Figure 1, we condition this prior on a single observation of flowing electric current, which leads to the magnetic field circling around the current. This usage of differential equations shows an extrapolation away from the data point in space and into other components.

The inhomogenous Maxwell equations are given by the operator matrix

$$
A := \begin{bmatrix}
0 & -\partial_z & \partial_y & \partial_t & 0 & 0 & 0 & 0 & 0 & 0 \\
\partial_z & 0 & -\partial_x & 0 & \partial_t & 0 & 0 & 0 & 0 & 0 \\
-\partial_y & \partial_x & 0 & 0 & 0 & \partial_t & 0 & 0 & 0 & 0 \\
0 & 0 & 0 & \partial_x & \partial_y & \partial_z & 0 & 0 & 0 & 0 \\
-\partial_t & 0 & 0 & 0 & -\partial_z & \partial_y & -1 & 0 & 0 & 0 \\
0 & -\partial_t & 0 & \partial_z & 0 & -\partial_x & 0 & -1 & 0 & 0 \\
0 & 0 & -\partial_t & -\partial_y & \partial_x & 0 & 0 & 0 & -1 & 0 \\
\partial_x & \partial_y & \partial_z & 0 & 0 & 0 & 0 & 0 & 0 & -1
\end{bmatrix}
$$

applied to three components of the electric field, three components of the magnetic (pseudo) field, three components of electric current, and one component of electric flux. We have set all constants to 1.

Using Gröbner bases, one computes the right kernel

$$
B := \begin{bmatrix}
\partial_x & \partial_t & 0 & 0 \\
\partial_y & 0 & \partial_t & 0 \\
\partial_z & 0 & 0 & \partial_t \\
0 & 0 & \partial_z & -\partial_y \\
0 & -\partial_z & 0 & \partial_x \\
0 & \partial_y & -\partial_x & 0 \\
-\partial_t\partial_x & \partial_y^2 + \partial_z^2 - \partial_t^2 & -\partial_y\partial_x & -\partial_z\partial_x \\
-\partial_t\partial_y & -\partial_y\partial_x & \partial_x^2 + \partial_z^2 - \partial_t^2 & -\partial_z\partial_y \\
-\partial_t\partial_z & -\partial_z\partial_x & -\partial_z\partial_y & \partial_x^2 + \partial_y^2 - \partial_t^2 \\
\partial_x^2 + \partial_y^2 + \partial_z^2 & \partial_t\partial_x & \partial_t\partial_y & \partial_t\partial_z
\end{bmatrix}
$$

of $A$ and verifies that it parametrizes the set of solutions of the inhomogeneous Maxwell equations.

For the demonstration in Figure 1 we assume squared exponential covariance functions and a zero mean function for four uncorrelated parametrising functions (electric potential and magnetic potentials).

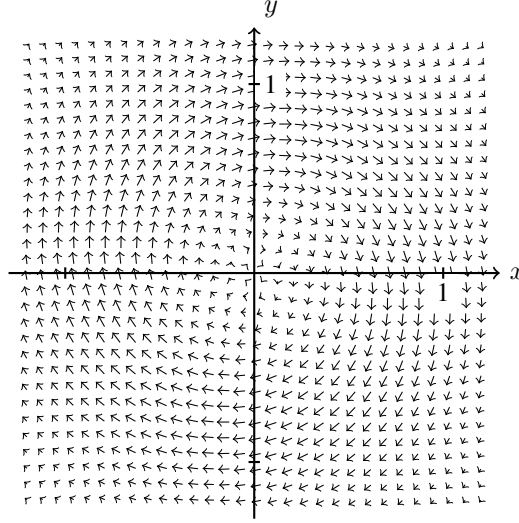

Figure 1: We condition the prior on solutions of Maxwell's equations from Example 4.1 on an electric current in $z$-direction and zero electric flux at the origin $x = y = z = t = 0$. The diagram shows the mean posterior magnetic field in the $(z, t) = (0, 0)$-plane. As expected by the right hand rule, it circles around the point with electric current.

**Example 4.2.** We consider the time-varying control system $\partial_t x(t) = t^3 u(t)$ from [34, Example 1.5.7] over the one-dimensional Weyl algebra $R = \mathbb{R}(t)\langle \partial_t \rangle$.

This system, given by the matrix $A := \begin{bmatrix} \partial_t & -t^3 \end{bmatrix}$, is parametrizable by

$$B = \begin{bmatrix} 1 \\ \frac{1}{t^3} \partial_t \end{bmatrix} .$$

For a parametrizing functions with squared exponential covariance functions $k(t_1, t_2) = \exp(-\frac{1}{2}(t_1 - t_2)^2)$ and a zero mean function, the covariance function for $(x, u)$ is

$$\begin{bmatrix} 1 & \frac{t_1 - t_2}{t_2^3} \\ \frac{t_2 - t_1}{t_1^3} & \frac{(t_2 - t_1 - 1)(t_1 - t_2 - 1)}{t_2^3 t_1^3} \end{bmatrix} \exp\left(-\frac{1}{2}(t_1 - t_2)^2\right) .$$

For a demonstration of how to observe resp. control such a system see Figures 2 resp. 3.

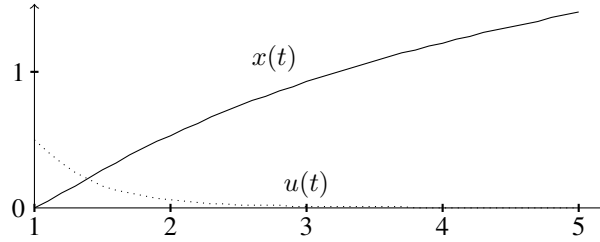

Figure 2: The state function $x(t)$ of the system in Example 4.2 can be influenced by assigning an input function $u(t)$. E.g., leaving the state $x(t)$ unspecified except for the boundary condition $x(1) = 0$ and fixing the input $u(t) = \frac{1}{t^4 + 1}$ for $t \in \{1, \frac{11}{10}, \frac{12}{10}, \ldots, 5\}$ leads to the above posterior means. This model yields $x(5) \approx 1.436537$, close to $\int_1^5 \frac{t^3}{t^4 + 1}\, dt \approx 1.436551$.

**Example 4.3.** We reproduce the well-known fact that divergence-free (vector) fields can be parametrized by the curl operator. This has been used in connection with Gaussian processes to model electric and magnetic phenomena [29, 50, 45]. The same algebraic computation also constructs a prior for tangent fields of a sphere.

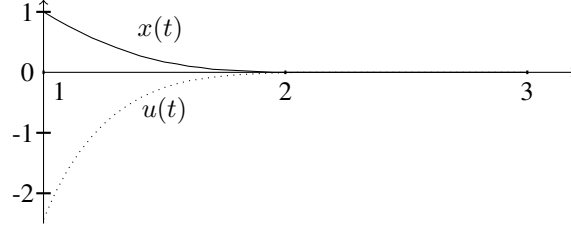

Figure 3: We control the system in Example 4.2 by specifying a desired behavior for the state $x(t)$ and letting the Gaussian process construct a suitable input $u(t)$, which is completely unspecified by us. Starting with $x(1) = 1$ we give $u(t)$ one time step to control $x(t)$ to zero, e.g., by setting $x(t) = 0$ for $t \in \{\frac{20}{10}, \frac{21}{10}, \ldots, 5\}$.

Let $R = \mathbb{Q}[x_1, x_2, x_3]$ resp. $R = \mathbb{Q}[\partial_1, \partial_2, \partial_3]$ be the polynomial ring in three indeterminates, which we can both interpret as the polynomial ring in the coordinates resp. in the differential operators. Consider the matrix $A = \begin{bmatrix} x_1 & x_2 & x_3 \end{bmatrix}$ representing the normals of circles centered around the origin resp. the divergence. The right kernel of $A$ is given by the operator

$$B = \begin{bmatrix} 0 & x_3 & -x_2 \\ -x_3 & 0 & x_1 \\ x_2 & -x_1 & 0 \end{bmatrix}$$

representing tangent spaces of circles centered around the origin resp. the curl, and these parametrize the solutions of $A$. A posterior mean field is demonstrated in Figure 4 when assuming equal covariance functions $k$ for 3 uncorrelated parametrizing functions and the covariance function for the tangent field is

$$\begin{bmatrix} y_1 y_2 + z_1 z_2 & -y_1 x_2 & -z_1 x_2 \\ -x_1 y_2 & x_1 x_2 + z_1 z_2 & -z_1 y_2 \\ -x_1 z_2 & -y_1 z_2 & x_1 x_2 + y_1 y_2 \end{bmatrix} \cdot k(x_1, y_1, z_1, x_2, y_2, z_2) \, .$$

We demonstrate how to compute $B$ and $A'$ for this example using Macaulay2 [16].

```
i1 : R=QQ[d1,d2,d3]
o1 = R
o1 : PolynomialRing
i2 : A=matrix{{d1,d2,d3}}
o2 = | d1 d2 d3 |
              1        3
o2 : Matrix R  <--- R
i3 : B = generators kernel A
o3 = {1} | -d2 0   -d3 |
{1} | d1  -d3 0   |
{1} | 0   d2  d1  |
              3        3
o3 : Matrix R  <--- R
i4 : A1 = transpose generators kernel transpose B
o4 = | d1 d2 d3 |
              1        3
o4 : Matrix R  <--- R
```

**Example 4.4.** We construct a prior for smooth tangent fields on the sphere without sources and sinks. We work in the third polynomial Weyl algebra $R = \mathbb{R}[x, y, z]\langle \partial_x, \partial_y, \partial_z \rangle$. I.e., we are interested in $\mathrm{sol}_A(\mathcal{F}) = \{v \in C^\infty(S^2, \mathbb{R}^3) | Av = 0\}$ for

$$A := \begin{bmatrix} x & y & z \\ \partial_x & \partial_y & \partial_z \end{bmatrix} \, .$$

The right kernel

$$B := \begin{bmatrix} -z\partial_y + y\partial_z \\ z\partial_x - x\partial_z \\ -y\partial_x + x\partial_y \end{bmatrix}.$$

can be checked to yield a parametrization of $\mathrm{sol}_{\mathcal{F}}(A)$ Assuming a squared exponential covariance functions $k$ for the parametrizing function, a demonstration can be found in Figure 4

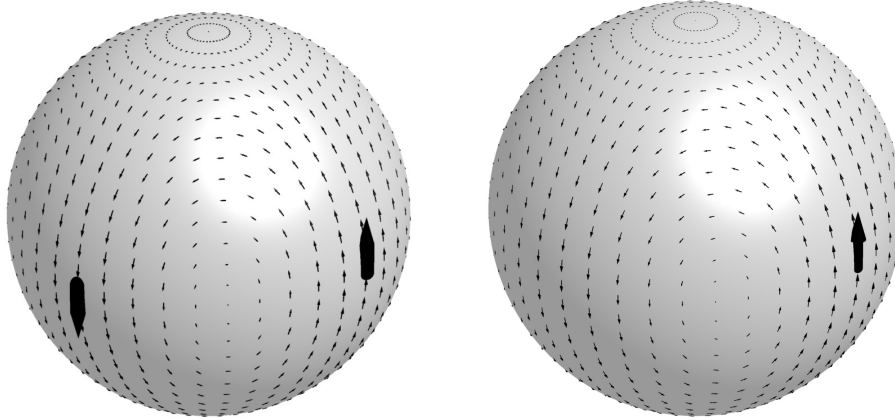

Figure 4: Taking the squared exponential covariance function for $k$ in Example 4.3 yields the left smooth mean tangent field on the sphere after conditioning at 4 evenly distributed points on the equator with two opposite tangent vectors pointing north and south each. The two visible of these four vectors are displayed significantly bigger. Conditioning the prior in Example 4.4 at 2 opposite points on the equator with tangent vectors both pointing north (displayed bigger) yields the right mean field.

## 5 Conclusion

The paper constructs multi-output Gaussian process priors, which adhere to linear operator equations. With these priors few observations yield a precise regression model with strong extrapolation capabilities (cf. Examples 4.1, 4.3, and 4.4). This construction is fully algorithmic and rather general, as it allows linear systems of differential equations with constant or variable coefficients, shift equations, or multiplications with variables. It could be applied to settings from physics (cf. Examples 4.1), geometric settings with potential applications in geomathematics and weather prediction (cf. Examples 4.1, 4.3, and 4.4), or to observe and control systems (cf. Example 4.2). The main restriction is that the solutions of the system of equations must be parametrizable.

The author hopes that the results can be generalized from parametrizable solution sets to the general case using a Monge parametrization (computable via the purity filtration [36, 35, 2]) and right hand sides [15]. It would also be interesting to apply to parameter estimation (cf. Example 4.2), boundary conditions [15], and to clarify the connection between the algebra, functional analysis, topology, and measure theory used in this paper. Finally, experimental results would be interesting which covariance function for the parametrizing functions is most suitable.

## Acknowledgments

The authors thanks M. Barakat, S. Gutsche, C. Kaus, D. Moser, S. Posur, and O. Wittich for discussions concerning this paper, W. Plesken, A. Quadrat, D. Robertz, and E. Zerz for introducing him to the algebraic background of this paper, S. Thewes for introducing him to Gaussian processes, and the authors of [24] for providing the starting point of this work. This work owes much to comments from anonymous reviewers.

## Footnotes

[1]They are the maximum entropy prior for finite mean and variance in the unknown behavior [22, 23].

[2]For notational simplicity, we refrain from using the phrases "almost surely" and "up to equivalence" in this paper, e.g. by assuming separability.

[3]The construction of covariance functions is applicable to kernels more generally.

[4]No major changes for polynomial, holonomic, or meromorphic coefficients.

[5]As finite dimensional vector spaces are reflexive, i.e. isomorphic to their bi-dual.

[6]More precisely, $A$ and $A'$ have the same row space.

[7]This standard form depends on choices, specifically a so-called monomial order.

[8]Sometimes even the mean function contains hyperparameters. These additional hyperparameters are usually not very expressive, compared to the non-parametric Gaussian process model.

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
