[Reviews · NeurIPS 2018]

Reviewer 1



The paper addresses the problem of constructing GP priors satisfying constraints that can be expressed as algebraic problems. They demonstrate how this is facilitated by the application of Groebner bases. They discuss some of the mathematical prorperties and show example applications. I will first address the biggest weakness of the paper: its clarity. From the presentation, it seems evident that the author is an expert in the field of computer algebra/algebraic geometry. It is my assumption that most members of the NIPS community will not have a strong background on this subject, me including. As a consequence, I found it very hard to follow Sect. 3. My impression was that the closer the manuscript comes to the core of algebraic geometry results, the less background was provided. In particular, I would have loved to see at least a proof idea or some more details/background on Thm. 3.1 and Cor. 3.2. Or maybe, the author could include one less example in the main text but show the entire derivation how to get from one concrete instance of A to right kernel B by manual computation? Also, for me the description in Sect. 2.4 was insufficient. As a constructive instruction, maybe drop one of the examples (R(del_t) / R[sigma_x]), but give some more background on the other? This problem of insufficient clarity cannot be explained by different backgrounds alone. In Sect. 3.2, the sentence "They are implemented in various computer algebra systems, 174 e.g., Singular [8] and Macaulay2 [16] are two well-known open source systems." appears twice (and also needs grammar checking). If the author could find a minimal non-trivial example (to me, this would be an example not including the previously considered linear differential operator examples) for which the author can show the entire computation in Sect. 3.2 or maybe show pseudo-code for some algorithms involving the Groebner basis, this would probably go a long way in the community. That being said, the paper's strengths are (to the best of this reviewer's knowledge) its originality and potential significance. The insight that Groebner bases can be used as a rich language to encode algebraic constraints and highlighting the connection to this vast background theory opens an entirely new approach in modelling capacities for Gaussian processes. I can easily imagine this work being the foundation for many physical/empirical-hybrid models in many engineering applications. I fully agree and applaud the rationale in lines 43-54! Crucially, the significance of this work will depend on whether this view will be adopted fast enough by the rest of the community which in turn depends on the clarity of the presentation. In conclusion: if I understood the paper correctly, I think the theory presented therein is highly original and significant, but in my opinion, the clarity should be improved significantly before acceptance, if this work should reach its full potential. However, if other reviewers have a different opinion on the level of necessary background material, I would even consider this work for oral presentation. Minor suggestions for improvements: - In line 75, the author writes that the "mean function is used as regression model" and this is how the author uses GPs throughout. However, in practice the (posterior) covariance is also considered as "measure of uncertainty". It would be insightful, if the author could find a way to visualize this for one or two of the examples the author considers, e.g., by drawing from the posterior process. - I am not familiar with the literature: all the considerations in this paper should also be applicable to kernel (ridge) regression, no? Maybe this could also be presented in the 'language of kernel interpolation/smoothing' as well? - I am uncertain about the author's reasoning on line 103. Does the author want to express that the mean is a sample from the GP? But the mean is not a sample from the GP with probability 1. Generally, there seems to be some inconsistency with the (algebraic) GP object and samples from said object. - The comment on line 158 "This did not lead to practical problems, yet." is very ominous. Would we even expect any problem? If not, I would argue you can drop it entirely. - I am not sure whether I understood Fig. 2 correctly. Am I correct that u(t) is either given by data or as one draw from the GP and then, x(t) is the corresponding resulting state function for this specified u? I'm assuming that Fig. 3 is done the other way around, right? --- Post-rebuttal update: Thank you for your rebuttal. I think that adding computer-algebra code sounds like a good idea. Maybe presenting the work more in the context of kernel ridge regression would eliminate the discussion about interpreting the uncertainty. Alternatively, if the author opts to present it as GP, maybe a video could be used to represent the uncertainty by sampling a random walk through the distribution. Finally, it might help to not use differential equations as expository material. I assume the author's rationale for using this was that reader might already a bit familiar with it and thus help its understanding. I agree, but for me it made it harder to understand the generality with respect to Groebner bases. My first intuition was that "this has been done". Maybe make they Weyl algebra and Figure 4 the basic piece? But I expect this suggestion to have high variance.

Reviewer 2



This paper addresses the interesting problem of creating multi-output kernels which produce GPs that have samples that satisfy some linear operator, like a partial differential equation. Kernel methods (both single and mullti-output) are hampered in particular by the simple inductive biases that kernels can encode. By embedding very strong constraints that fit reality, like physical laws described as a PDE, kernels can make much better predictions. This contribution will be gladly appreciated by the community. This paper's main contribution is a characterisation when a GP that satisfy a set of linear equations exists, and a constructive method for finding kernels that do. The main drawback of the paper is that it is not written in a particularly accessible way. Theorems are stated with little context. E.g. eq (1) can be informally derived by considering expectations of operators on f at particular input locations. Additionally, the core methodology used in the paper, Gröbner bases, is only cited. There is no discussion of computational complexity, or a basic overview of the steps of the algorithm. The examples in §4 seem correct, but the paper itself does not help much for a reader wanting to apply the method to their own problem. It could also be helpful for a reader to see an explicitly worked out covariance function for a simple problem. E.g. the kernel for example 4.2 could be easily reproduced. Also, the example could be a lot more illustrative if the paper was more explicit about what variables are being conditioned on, and what are being predicted in the GP. The reader is left to do a lot of work. Overall, the contribution is interesting and significant enough for a NIPS paper. Addendum in response to the rebuttal We thank the authors for their rebuttal. It is hard to address improvements to the clarity of the paper in the rebuttal, although it is encouraging that the authors acknowledge room for improvement as well. I believe my assessment of the paper after the rebuttal to be in line with my assessment after the rebuttal, and will therefore leave my scores unchanged. Regarding the choice of presenting Groebner bases as a black box, I have an additional (and personal) suggestion on how to improve clarity. Often the full mathematical machinery is not needed to convey the core modelling assumption which is relevant to the inductive bias of the method. For an 8 page conference paper, I believe that it is often clearer to introduce the constraints of the algorithm for a more specific case (or even a running example), than to try to introduce a general framework that relies on mathematical details. Sacrificing some generality for specificity can convey the goal of the method to more readers in an easier way. A more detailed mathematical treatment can be left to the appendix or a journal paper.

Reviewer 3



The paper is about building covariance functions such that the samples satisfy some differential equations. This can be seen as building more informative priors where some physical knowledge about the system is encoded in the covariance. Let A be a linear given differential operator that takes as input a vector of function and returns a vector of linear combinations of the input partial derivatives. The authors are interested in building a GP f satisfying Af=0. Their approach consists in using Grobner basis for finding an operator B such that ABg=0 for all g in a given space, and then to define f as f=Bg. Some care is provided to ensure that all the elements that are solution of Af=0 can actually be written as f=Bg. This is a very technical paper and I can not vouch for the correctness of the derivations but the authors did a great job at conveying the important ideas of the paper without burying them behind the technicality. For example, I really appreciated the discussion on the Existence of parametrization (sec 3.1). The overall idea to build more informative prior is particularly relevant to the community and I see this paper as an interesting, and general way to do so. There are already some kernel such that the samples satisfy differential equations in the literature but I guess the proposed method is the first one that guaranties that all possible solutions of the equations can be generated with the obtained kernel. In a nutshell, this appear to me as a good paper with a good attention to clarity and possibly a high significance. I guess the method used are classical in some communities but that they will be new to most of the Machine Learning community. I however do not have the background to check the correctness of the derivations and I hope one of the reviewer will be able to do that. I would have appreciated more details on the Grobner basis and on how to test if an operator B parameterize the set of solutions. Finally, I have identified the following typos: - "geomathematic" is missing an "e" (two occurences) - line 7: missing space - l. 165 echolon -> echelon - sentence line 173 is repeated line 178. ################### After rebuttal: The author's feedback is helpful to understand why they chose to present their paper this way. The minor changes that have been made should result in an improvement of the clarity of the manuscript, but overall the paper seems to remain the same. I thus choose to keep my initial recommendation.